# Validation of prognostic scores predicting mortality in acute liver decompensation or acute-on-chronic liver failure: A Thailand multicenter study

**Tongluk Teerasarntipan**[1], **Kessarin Thanapirom**[1], **Sakkarin Chirapongsathorn**[2], **Tanita Suttichaimongkol**[3], **Naichaya Chamroonkul**[4], **Chalermrat Bunchorntavakul**[5], **Sith Siramolpiwat**[6], **Siwaporn Chainuvati**[7], **Abhasnee Sobhonslidsuk**[8], **Apinya Leerapun**[9], **Teerha Piratvisuth**[4], **Wattana Sukeepaisarnjaroen**[3], **Tawesak Tanwandee**[7], **Sombat Treeprasertsuk**[1]*

1 Department of Medicine, Division of Gastroenterology, Faculty of Medicine, Chulalongkorn University, and Thai Red Cross, Bangkok, Thailand, 2 Division of Gastroenterology and Hepatology, Phramongkutklao Hospital and College of Medicine, Royal Thai Army, Bangkok, Thailand, 3 Division of Gastroenterology and Hepatology, Department of Medicine, Faculty of Medicine, Khon Kaen University, Khon Kaen, Thailand, 4 Division of Gastroenterology and Hepatology, Department of Medicine, Faculty of Medicine, Prince of Songkla University, Songkhla, Thailand, 5 Division of Gastroenterology and Hepatology, Department of Medicine, Rajavithi Hospital, College of Medicine, Rangsit University, Bangkok, Thailand, 6 Division of Gastroenterology, Department of Internal Medicine, Faculty of Medicine, Thammasat University, Pathumthani, Thailand, 7 Division of Gastroenterology, Department of Medicine, Siriraj Hospital, Bangkok, Thailand, 8 Ramathibodi Hospital, Faculty of Medicine, Mahidol University, Bangkok, Thailand, 9 Department of Internal Medicine, Chiang Mai University, Chiang Mai, Thailand

* battan5410@gmail.com, sombat.t@chula.ac.th

## Abstract

### Background & objectives

Cirrhosis patients with worsening of the liver function are defined as acute decompensation (AD) and those who develop extrahepatic organ failure are defined as acute-on-chronic liver failure (ACLF). Both AD and ACLF have an extremely poor prognosis. However, information regarding prognostic predictors is still lacking in Asian populations. We aimed to identify prognostic factors for 30-day and 90-day mortality in cirrhosis patients who develop AD with or without ACLF.

### Methods

We included 9 tertiary hospitals from Thailand in a retrospective observational study enrolling hospitalized cirrhosis patients with AD. ACLF was diagnosed according to the EASL-CLIF criteria, which defined as AD patients who have kidney failure or a combination of at least two non-kidney organ failure. Outcomes were clinical parameters and prognostic scores associated with mortality evaluated at 30 days and 90 days.

### Results

Between 2015 and 2020, 602 patients (301 for each group) were included. The 30-day and 90-day mortality rates of ACLF vs. AD were 57.48% vs. 25.50% (p<0.001) and 67.44% vs.

**Data Availability Statement:** All relevant data are within the paper and its Supporting Information files.

**Funding:** This study received financial support by the Thai Association for the Study of the Liver (THASL) and Fatty Liver Foundation, Division of Gastroenterology, Department of Medicine, Faculty of Medicine, Chulalongkorn University, Bangkok, Thailand. The funders had no role in study design, data collection and analysis, decision to publish, or preparation of the manuscript.

**Competing interests:** The authors have declared that no competing interests exist.

32.78% (p<0.001), respectively. For ACLF patients, logistic regression analysis adjusted for demographic data, and clinical information showed that increasing creatinine was a predictor for 30-day mortality (p = 0.038), while the CLIF-C OF score predicted both 30-day (p = 0.018) and 90-day (p = 0.037) mortalities, achieving the best discriminatory power with AUROCs of 0.705 and 0.709, respectively. For AD patients, none of the parameters was found to be significantly associated with 30-day mortality, while bacterial infection, CLIF-AD score and Child-Turcotte-Pugh score were independent parameters associated with 90-day mortality, with p values of 0.041, 0.024 and 0.024. However, their predictive performance became nonsignificant after adjustment by multivariate regression analysis.

## Conclusions

Regarding Thai patients, the CLIF-C OF score was the best predictor for 30-day and 90-day mortalities in ACLF patients, while appropriate prognostic factors for AD patients remained inconclusive.

## Introduction

Patients with cirrhosis who present with acute deterioration of their liver functions, including recent onset of ascites, gastrointestinal (GI) bleeding, hepatic encephalopathy (HE), bacterial infections, or any combination of these symptoms, are traditionally termed "cirrhosis with acute decompensation (AD)" patients [1]. Patients who have AD and develop multiorgan failure, which is associated with high short-term mortality, are recognized as having acute-on-chronic liver failure (ACLF) [2]. Recently, we conducted a single-center study and reported 30-day mortality rates (MRs) of 13.3% and 43.7%-51% in patients with AD and ACLF, respectively [3], indicating a worrisome healthcare burden; however, epidemiological information from a large study that could precisely estimate disease burden in Thailand is still lacking.

The early detection of poor prognostic parameters and the triage of patients with a high risk of death for special care is a practical strategy to improve clinical outcomes in resource-limited countries. Previous studies found that the prognoses of AD and ACLF patients varied according to the cirrhosis etiology, precipitating factors and geographic location (Eastern vs. Western countries) [4, 5]. Therefore, the aim of this national multicenter study was to identify prognostic factors for short-term mortality in hospitalized cirrhosis patients who develop AD with or without ACLF. In addition, we aimed to validate the CLIF-C OF score and CLIF-C ACLF score in our Asian population.

## Materials and methods

### Study design

The ACLF Thailand multicenter (ACLF-TM) study group enrolled 9 tertiary hospitals that are representatives of hospitals in major regions of Thailand, including the central region, northern region, northeastern region, southern region and Bangkok. We performed a retrospective observational study collecting information from January 2019 to June 2020. This study followed the STROBE guideline during the preparation of manuscript. The individual consent for retrospective analysis was waived and all subjects participated in prospective data collection gave their informed consent before study enrollment. The study protocol was approved by the

ethics committee of the Faculty of Medicine, Chulalongkorn University (IRB number 330/59). This study conformed to the ethical guidelines of the Helsinki Declaration.

We enrolled cirrhosis patients aged 18 years or older who were hospitalized due to AD of the liver from January 2015 to June 2020. The diagnosis of cirrhosis was based on one of the following: 1) a combination of clinical, biochemical and radiologic studies or 2) histopathology of stage 4 fibrosis. We excluded patients if they had one of the following conditions: 1) chronic kidney disease according to the Kidney Disease Outcomes Quality Initiative (KDOQI) guideline definitions (glomerular filtration rate <60 mL/min/1.73 m$^2$ for three months or kidney damage (functional or structural abnormalities) for equal to or greater than three months); 2) hospitalization due to schedule for treatment or procedure; 3) comorbidity of severe chronic extrahepatic disease; 4) receiving immunosuppressive drugs except severe alcoholic hepatitis; 5) pregnancy; and 6) concurrent hepatocellular carcinoma regardless of disease activity. We defined bacterial infection by one of the following conditions: 1) detection of positive bacterial culture in blood or any clinically suspicious organ 2) clinically and laboratory compatible with sepsis or spontaneous bacterial peritonitis with-or without of microbiologically documented infection. The study eligibility of patients from individual study centers was considered by gastroenterologists whose major professional activities were clinical practice, teaching and conducting research in hepatology for at least two years.

The participants were categorized into the AD or ACLF group by clinician evaluation during hospitalization. AD of the liver was defined as the development of ascites, HE, GI hemorrhage, bacterial infection or a combination of these symptoms and not meeting the ACLF criteria [6]. The diagnosis of ACLF was based on the CANONIC study and the definition by the European Association for the Study of the Liver–Chronic Liver Failure (EASL-CLIF) consortium, which identified ACLF as organ failures of the liver, coagulation, kidney, circulation, lungs and cerebral systems [2]. Patients who had single kidney failure or single nonrenal organ failure with the presence of decreased kidney function (creatinine 1.5 to 1.9 mg/dL) or mild-to-moderate HE and more than two organ failures received a diagnosis of ACLF. Definitions of organ failure were based on the CLIF-SOFA score [2]. Patients who presented with AD features and developed ACLF during admission were categorized into the ACLF group. Demographic data, laboratory profiles and clinical presentations were collected at the date of admission for the AD group and at the date of ACLF diagnosis for the ACLF group. We used a web-based calculator (https://www.efclif.com/) as a tool for ACLF or AD diagnosis and ACLF grading and calculated other prognostic models, including the CLIF-C organ failure (CLIF-C OF) score for both AD and ACLF patients, the CLIF-C ACLF score for ACLF patients, and the CLIF-AD score for AD patients. We also calculated the Child-Turcotte-Pugh (CTP) score, Albumin-Bilirubin (ALBI) score, model for end-stage liver disease (MELD) score, and MELD-Na score to validate their prediction accuracies. The primary outcomes were mortality rates at 30 days and 90 days after study enrollment. All information was collected using electronic medical records. To avoid possible selection bias, participants information, classification and exclusion were validated by authors from different study centers.

## Statistical analysis

The sample size of calculated to estimate an infinite population proportion considering a mortality of 22.1% based on the ACLF grade 1 mortality rate from the CANONIC study [2] and it was found that a minimum of 340 patients were required to achieve this observational result with a power of 80 percent and 0.05 type 1 error. With additional 10% lost to follow-up rate, a total of 374 participants were required in this study. Categorical variables are described as counts and percentages and were compared using the chi-square test or Fisher's exact test.

Continuous variables are presented as the means; for comparisons between groups, a two-tailed independent sample t test was used for continuous variables with a normal distribution, and the Mann–Whitney (Wilcoxon rank) test was used for those with a nonnormal distribution. At the end of follow-up period, survival analysis between patients with different liver decompensation severity was performed using Kaplan-meier curve and compared between group by log rank test. The relationship between each predicted variable and mortality was initially assessed by logistic regression analysis. Significant parameters were subsequently included in multivariate Cox regression analysis (forward-stepwise model). The odds ratio (OR), calculated by logistic regression analysis and the Mantel–Haenszel method, was used to estimate the degree of each prognostic factor. Receiver operating characteristic (ROC) curves and the area under the ROC curve (AUROC) were estimated to evaluate the ability of the AD and ACLF prognostic scores to predict short-term mortality. Regarding missing data management, we used listwise deletion or complete case analysis. All tests were 2-sided, and the adopted p value for the significance level was < 0.05. SPSS version 23.0 was used for statistical analysis.

## Results

We recruited 638 patients in the study cohort. From that, we excluded 36 patients due to having concurrent hepatocellular carcinoma or other cancers detected by abdominal imaging during follow up. A total of 602 participants were enrolled in our study, including 301 patients with AD and 301 patients with ACLF. There were 48 participants (7.9%) retrospectively recruited after AD or ACLF occurred but the 90-day evaluation, we prospectively collected their data after the onset of liver decompensation events. The mean age was 58.4±14.2 years. Most participants (n = 396, 65.7%) were male. There was no difference in mean age or sex between the groups. The proportions of chronic liver insults were as follows: alcohol (n = 310, 51.4%), chronic hepatitis C virus (HCV) (n = 120, 19.9%), chronic hepatitis B virus (HBV) (n = 113, 18.7%) and nonalcoholic fatty liver disease (NAFLD) (n = 55, 9.1%), which were not different between the two groups, except for cryptogenic cirrhosis (ACLF 10.3% vs. AD 4.97%, p = 0.014). Two major acute insults were bacterial infection, which was more common in the ACLF group (59.14% vs. 47.13%, p<0.001), and GI bleeding, which was more common in the AD group (47.13% vs. 21.26%, p<0.001). Regarding laboratory profiles on admission, in addition to parameters from the CLIF-SOFA score (bilirubin, INR, and creatinine), ACLF patients had higher levels of serum markers of inflammation (white blood cell count, % neutrophils, lactate and acidosis) than AD patients. Serum albumin and other liver function tests were not different between the two groups (Tables 1 and 2).

ACLF patients had higher mortality rates than AD patients at both 30 days (57.48% vs. 25.50%, p<0.001, OR 3.96, 95% CI: 2.80–5.61) and 90 days (67.44% vs. 32.78%, p<0.001, OR 4.70, 95% CI: 3.27–6.74). Six patients (1.99%), who were currently on the waiting list before the onset of ACLF, underwent liver transplantation within 90 days after ACLF diagnosis. The numbers of patients with each ACLF grade were not significantly different (92, 106 and 103 for ACLF grades 1, 2 and 3, respectively, p = 0.404). Patients with a higher ACLF grade had higher mortality rates at both 30 days (34.78%, 54.72% and 77.67% for ACLF grades 1, 2 and 3, respectively, p<0.001) and 90 days (48.91%, 66.04% and 85.44% for ACLF grades 1, 2 and 3, respectively, p<0.001) (Fig 1). Kaplan-Meier curve showed significant different in survival between each group of liver decompensation severity, p<0.001. Log rank test results between AD vs. ACLF groups, ACLF grade 1 vs. ACLF grade 2, and ACLF grade 2 vs. ACLF grade 3 were p<0.001, p = 0.002, and p<0.001, respectively (Fig 2). Regarding non-survivor patients, median time to death (95% confidence interval) from AD, ACLF grade 1, ACLF grade 2, and

**Table 1. Baseline characteristics of patients with AD and ACLF.**

| Parameters (Mean±SD.) | AD (n = 301) | ACLF (n = 301) | p value |
|---|---|---|---|
| Age, years | 58.07±13.08 | 58.87±14.73 | 0.488 |
| Male (n, %) | 198 (65.8%) | 198 (65.8%) | 1.000 |
| Sodium (mEq/L) | 133.90±6.20 | 132.40±7.97 | 0.010 |
| Hemoglobin (g/L) | 9.81±4.37 | 9.60±2.47 | 0.460 |
| WBC (x10$^9$/L) | 9.71±6.58 | 14.19±16.69 | <0.001 |
| Platelets (x10$^9$/L) | 130.49±88.62 | 133.10±91.00 | 0.721 |
| INR | 1.53±0.37 | 2.27±1.23 | <0.001 |
| Creatinine (mg/dL) | 0.95±0.35 | 2.10±1.54 | <0.001 |
| Potassium (mEq/L) | 4.47±8.33 | 4.28±1.08 | 0.713 |
| Bicarbonate (mEq/L) | 21.15±6.51 | 17.36±6.32 | <0.001 |
| Total bilirubin (mg/dL) | 4.51±5.75 | 10.87±10.51 | <0.001 |
| AST (U/L) | 167.71±492.81 | 218.95±376.87 | 0.152 |
| ALT (U/L) | 71.38±185.98 | 96.83±194.40 | 0.101 |
| ALP (U/L) | 147.48±87.91 | 163.34±133.89 | 0.086 |
| Albumin (g/dL) | 2.88±1.94 | 2.65±2.11 | 0.165 |
| Lactate (mg/dL) | 3.56±3.33 | 6.48±6.40 | <0.001 |
| MELD score | 16.71±6.18 | 27.54±8.37 | <0.001 |
| MELD-Na score | 18.72±8.16 | 28.92±8.91 | <0.001 |
| CTP score | 8.95±2.30 | 11.47±3.11 | <0.001 |

AST, aspartate aminotransferase; ALT, alanine aminotransferase; ALP, alkaline phosphatase; MELD, Model for End-Stage Liver disease; CTP, Child-Pugh-Turcotte

ACLF grade 3 were 12.00 (3.86–20.14) days, 15.00 (8.66–21.34) days, 11.00 (9.22–12.78) days, and 5.00 (3.37–6.63) days, respectively.

**Table 2. Baseline liver conditions and precipitating factors of liver decompensation.**

| | AD (number, %) | ACLF (number, %) | p value | OR (95%CI) |
|---|---|---|---|---|
| Presence of diabetes mellitus | 74 (30.7%) | 45 (20.4%) | 0.014 | 0.58 (0.38–0.88) |
| **Cause of chronic liver disease** | | | | |
| Alcohol consumption | 154 (50.99%) | 156 (51.83%) | 0.871 | 0.97 (0.70–1.33) |
| HCV infection | 68 (22.52%) | 52 (17.28%) | 0.467 | 0.86 (0.57–1.29) |
| HBV infection | 53 (17.55%) | 60 (19.93%) | 0.126 | 1.39 (0.93–2.08) |
| Non-alcoholic fatty liver | 34 (11.26%) | 21 (6.98%) | 0.089 | 0.59 (0.34–1.04) |
| Cryptogenic | 15 (4.97%) | 31 (10.30%) | 0.014 | 2.20 (1.16–4.16) |
| Other cause* | 9 (2.98%) | 16 (5.32%) | 0.159 | 1.83 (0.80–4.20) |
| **Precipitating factor** | | | | |
| Bacterial infection | 123 (40.73%) | 178 (59.14%) | <0.001 | 2.11 (1.52–2.91) |
| Gastrointestinal bleeding | 123 (40.73%) | 64 (21.26%) | <0.001 | 0.39 (0.27–0.56) |
| Alcohol consumption | 20 (6.62%) | 32 (10.63%) | 0.084 | 1.68 (0.94–3.01) |
| Unidentified | 18 (5.96%) | 18 (5.98%) | 1.000 | 1.00 (0.51–1.97) |
| Other cause** | 22 (7.28%) | 20 (6.64%) | 0.873 | 0.91 (0.48–1.70) |

HCV, hepatitis C virus; HBV, hepatitis B virus

*Autoimmune hepatitis, primary biliary cholangitis, biliary atresia, Wilson's disease

**HBV reactivation, autoimmune hepatitis flare-up, drug-induced liver injury, Wilson's disease flare-up

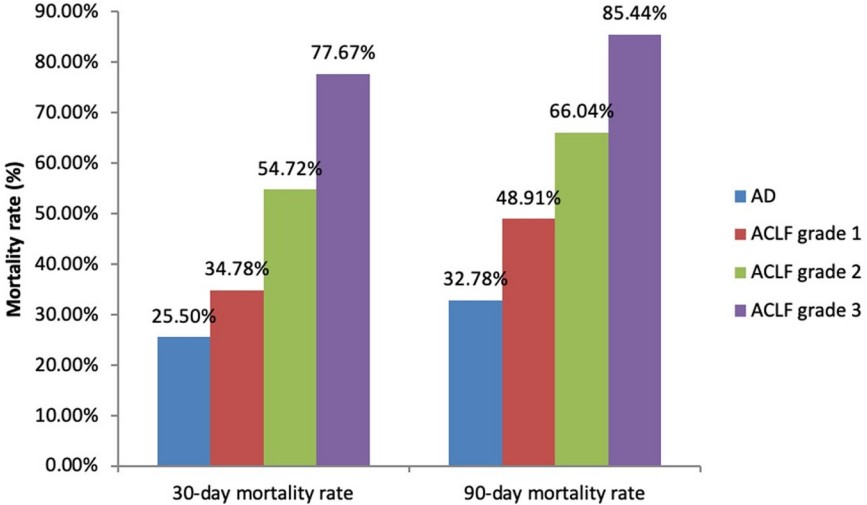

**Fig 1. Comparison of short-term mortality rates based on liver decompensation severity.**

For ACLF patients, the CLIF-C OF score and CLIF-C ACLF score had fair accuracies for predicting 30-day mortality, with AUROCs of 0.705 and 0.693, respectively, and 90-day mortality, with AUROCs of 0.709 and 0.705, while the CTP score, ALBI score, MELD score and MELD-Na score poorly predicted short-term mortality rates, with AUROCs of 0.599–0.642. Regarding AD patients, none of the scoring systems had good accuracy in predicting 30-day mortality (AUROCs of 0.451–0.580, p > 0.05). Regression analysis showed that the CTP score and CLIF-C AD score were associated with 90-day mortality (p = 0.024, both), but their accuracies were unsatisfactory, with AUROCs of 0.596 and 0.588, respectively (Fig 3 and S1 Table).

Univariate analysis showed that among ACLF patients, age, sex, presence of type 2 diabetes mellitus (T2DM) and bacterial infection did not predict short-term death, whereas creatinine, INR, bilirubin, lactate, bicarbonate, albumin, ACLF grading, and prognostic models (CLIF-C

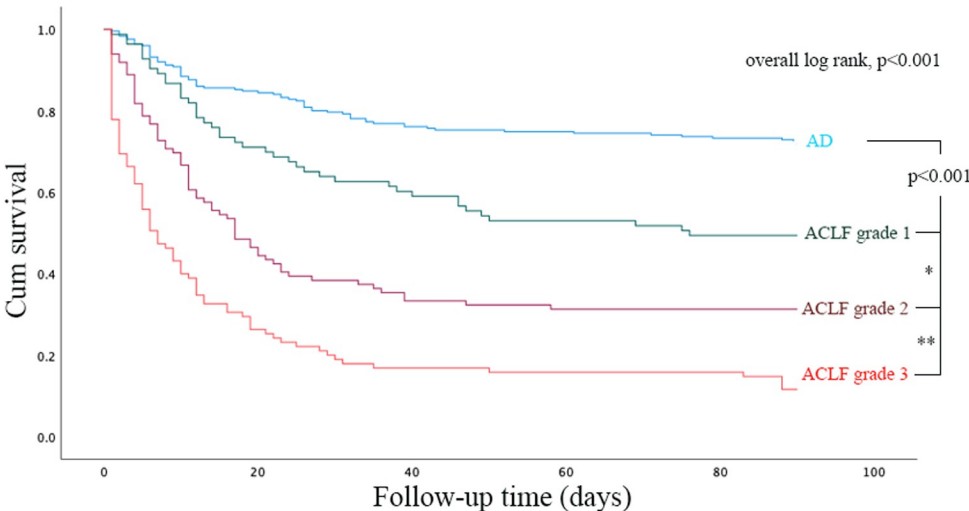

**Fig 2. Survival based on liver decompensation severity.** By overall, log rank tests showed survival difference between groups, p<0.001. Log rank test results between AD vs. ACLF groups, ACLF grade 1 vs. ACLF grade 2, and ACLF grade 2 vs. ACLF grade 3 were p<0.001, p = 0.002*, and p<0.001**, respectively.

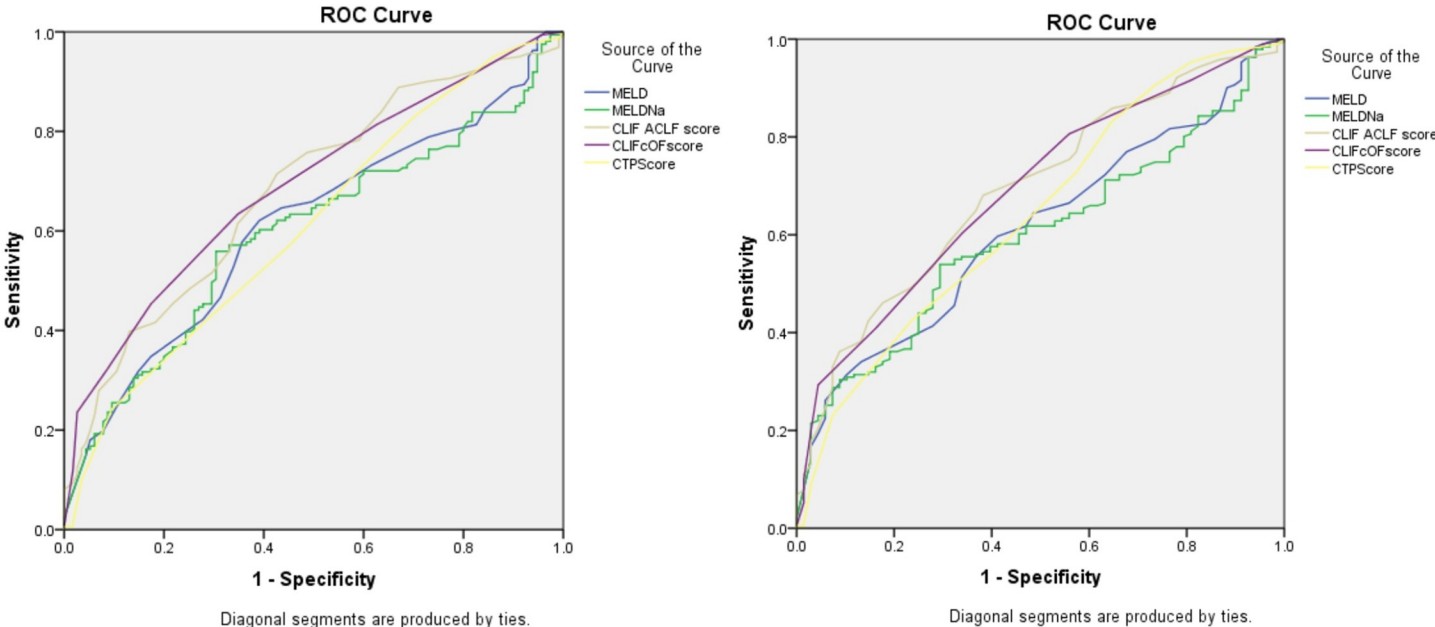

**Fig 3. Diagnostic performance of each prognostic score in predicting short-term mortality in ACLF patients.** a. 1-Month mortality prediction value. b. 3-Month mortality prediction value.

OF, CLIF-C ACLF, ALBI, MELD, MELD-Na, and CTP scores) were parameters associated with both 30-day and 90-day mortality rates. Multivariate analysis of standard prognostic scores found that CLIF-C OF scores were associated with 30-day and 90-day mortality, while CLIF-C ACLF scores was associated with 30-day mortality. (S2 Table) Subsequently, we performed multivariate analysis combining clinical parameters with selected prognostic scores. We found that serum creatinine and the CLIF-C OF score remained statistically significant for 30-day mortality prediction, with adjusted odds ratios (aORs) of 1.27 (95% CI: 1.01–1.59, p value = 0.038) and 1.44 (95% CI: 1.07–1.94, p value = 0.018), respectively, and the CLIF-C OF score remained the only predictor for 90-day mortality, with an aOR of 1.48 (95% CI: 1.02–2.13, p value = 0.037) (Tables 3 and S3).

Regarding prognostic predictors among AD patients, none of the initial parameters, including demographic background, bacterial infection, biochemistry profiles and prognostic models, were found to be potential 30-day prognostic predictors. For 90-day mortality, bacterial infection was an independent poor prognostic clinical parameter (p value = 0.049, aOR 1.65 (95% CI 1.00–2.73)). Of all prognostic scores, parameters associated with mortality were CLIF-AD score (aOR 1.14, 95% CI: 1.02–1.27, p value = 0.024) and CTP score (aOR 1.03, 95% CI: 1.00–1.06, p value = 0.024) (Tables 4 and S4). Unfortunately, all of the above parameters became statistically nonsignificant in predicting 90-day mortality after adjustment in multivariate regression analysis; aOR 1.47, 95% CI: 0.87–2.50, p value = 0.154 for bacterial infection, aOR 1.02, 95% CI: 1.00–1.05, p value 0.112 for CLIF-AD score and aOR 1.12, 95%CI: 1.00–1.26, p value = 0.055 for CTP score.

## Discussion

Currently, the concept of ACLF diagnosis and management has not been well recognized worldwide. To raise awareness to physicians, epidemiological information and outcomes are needed for a better understanding in each country. Only a few studies regarding ACLF issues

**Table 3. Predictive factors for mortality among patients with ACLF.**

| Factors | 30-Day mortality | | | | 90-Day mortality | | | |
|---|---|---|---|---|---|---|---|---|
| | Univariate analysis | | Multivariate analysis | | Univariate analysis | | Multivariate analysis | |
| | p value | OR (95%CI) | p value | OR (95%CI) | p value | OR (95%CI) | p value | OR (95%CI) |
| Bacterial infection | 0.654 | 0.90 (0.56–1.44) | | | 0.595 | 1.16 (0.68–1.97) | | |
| ACLF grade | <0.001 | 2.06 (1.77–2.41) | 0.393 | 0.76 (0.40–1.43) | <0.001 | 2.25 (1.89–2.68) | 0.454 | 0.75 (0.36–1.58) |
| Biochemical profiles | | | | | | | | |
| WBC | 0.055 | 1.00 (1.00–1.00) | | | 0.075 | 1.00 (1.00–1.00) | | |
| INR | <0.001 | 2.01 (1.58–2.56) | 0.524 | 0.91 (0.69–1.20) | <0.001 | 2.00 (1.53–2.61) | 0.075 | 0.76 (0.57–1.03) |
| Creatinine | <0.001 | 1.48 (1.26–1.73) | 0.038 | 1.27 (1.01–1.59) | <0.001 | 1.61 (1.33–1.94) | 0.063 | 1.30 (0.99–1.72) |
| Bicarbonate | 0.006 | 0.96 (0.93–0.99) | 0.838 | 0.99 (0.94–1.05) | 0.013 | 0.96 (0.94–0.99) | 0.262 | 1.04 (0.97–1.12) |
| Total bilirubin | <0.001 | 1.04 (1.02–1.06) | 0.596 | 1.01 (0.98–1.05) | <0.001 | 1.04 (1.02–1.06) | 0.462 | 1.02 (0.98–1.06) |
| ALT | 0.001 | 1.00 (1.00–1.00) | | | 0.009 | 1.00 (1.00–1.00) | | |
| Albumin | 0.006 | 0.70 (0.55–0.91) | 0.404 | 0.89 (0.68–1.17) | 0.061 | 0.89 (0.79–1.01) | 0.257 | 0.75 (0.46–1.23) |
| Lactate | <0.001 | 1.09 (1.04–1.14) | 0.415 | 1.03 (0.96–1.10) | 0.013 | 1.06 (1.01–1.12) | 0.235 | 1.05 (0.97–1.15) |
| Prognostic scores | | | | | | | | |
| CLIF-C OF score | <0.001 | 1.43 (1.32–1.55) | 0.018 | 1.44 (1.07–1.94) | <0.001 | 1.47 (1.35–1.60) | 0.037 | 1.48 (1.02–2.13) |
| CLIF-C ACLF | <0.001 | 1.07 (1.05–1.11) | 0.394 | 1.02 (0.98–1.06) | <0.001 | 1.08 (1.05–1.12) | 0.576 | 1.01 (0.97–1.06) |
| CTP score | <0.001 | 1.22 (1.14–1.30) | | | <0.001 | 1.29 (1.20–1.38) | | |
| MELD score | <0.001 | 1.07 (1.05–1.09) | | | <0.001 | 1.08 (1.06–1.11) | | |
| ALBI score | <0.001 | 3.19 (1.67–6.08) | | | 0.08 | 1.82 (0.93–3.55) | | |

INR, international normalized ratio; ALT, alanine aminotransferase; CTP, Child-Pugh-Turcotte; MELD, Model for End-Stage Liver Disease; ALBI, Albumin-Bilirubin

have been conducted in Asian populations [3, 7, 8]. This study is a multicenter study comprised of 9 university hospitals from all six geographical regions across the country. Our findings clearly showed the clinical presentations and short-term outcomes of this disease.

Previous epidemiological studies found that HBV infection is the main cause of cirrhosis in the Asia-Pacific region [9]; however, our study showed that alcoholic cirrhosis was the most

**Table 4. Predictive factors by univariate analysis for mortality among patients with AD without ACLF.**

| Clinical parameters | 30-Day mortality | | 90-Day mortality | |
|---|---|---|---|---|
| | p value | OR (95%CI) | p value | OR (95%CI) |
| Bacterial infection | 0.271 | 1.34 (0.80–2.67) | 0.049 | 1.65 (1.00–2.73) |
| White blood cell count | 0.284 | 1.00 (1.00–1.00) | 0.225 | 1.00 (1.00–1.00) |
| INR | 0.395 | 1.34 (0.68–2.62) | 0.179 | 1.56 (0.82–2.99) |
| Creatinine | 0.270 | 1.51 (0.73–3.15) | 0.116 | 1.75 (0.87–3.50) |
| Total bilirubin | 0.599 | 0.99 (0.94–1.04) | 0.780 | 0.99 (0.95–1.04) |
| ALT | 0.289 | 1.00 (1.00–1.00) | 0.301 | 1.00 (1.00–1.00) |
| Albumin | 0.373 | 0.90 (0.70–1.14) | 0.107 | 0.72 (0.48–1.07) |
| Lactate | 0.984 | 1.00 (0.89–1.12) | 0.234 | 0.93 (0.83–1.05) |
| CLIF-AD score | 0.194 | 1.02 (0.99–1.04) | 0.024 | 1.03 (1.00–1.06) |
| CLIF-OF score | 0.240 | 1.14 (0.92–1.42) | 0.252 | 1.13 (0.92–1.39) |
| CTP score | 0.102 | 1.10 (0.98–1.23) | 0.024 | 1.14 (1.02–1.27) |
| MELD score | 0.551 | 1.01 (0.97–1.06) | 0.076 | 1.04 (1.00–1.08) |

INR, international normalized ratio; ALT, alanine aminotransferase; CTP, Child-Pugh-Turcotte; MELD, Model for End-Stage Liver Disease

common in patients who developed either AD or ACLF, followed by HCV and HBV infection, which corresponded with findings from other ACLF studies [10]. For acute precipitating factors, bacterial translocation, a potent precipitating factor for systemic inflammation, particularly pathogen-associated molecular patterns (PAMPs) and damage-associated molecular patterns (DAMPs) [11], which are the main pathophysiology of ACLF, precipitated more ACLF than AD and was undoubtedly the most common insult in the ACLF group, as shown in the current findings and those of most other studies [10]. We found that GI bleeding and alcohol consumption were also leading triggers of AD, particularly GI bleeding, which accounted for 40.7% of patients with AD and 21.6% of patients with ACLF. These rates were higher than those in most other studies [12–14]. Because both factors are preventable, it might be inferred that primary prevention care for alcohol cirrhotic patients remains an important issue to address, and we suggest the promotion of active variceal surveillance and encourage medication and/or endoscopic therapy to be implemented to improve health care.

Although the ACLF 30-day mortality rate in the current study was slightly higher than that in most studies from Asia (57.48% vs. 40.98%-51.7%), our 90-day mortality rate was relatively similar to that in most Asian studies (67.44% vs. 49.2%-68.0%) [15–18]. Our mortality rates were higher than those in studies from the West, with reported 30-day and 90-day mortality rates of 33.0%-41.6% and 51.0%-57.4%, respectively [2, 14, 19–22]. As the proportion of cirrhosis etiologies and precipitating factors in our study resembled those in Western countries, the prognoses were in accordance with those in other Asian populations, which might suggest that demographic backgrounds and types of liver insult might have some effect on patient survival but are not very influential. Our findings suggested that the number and severity of organ dysfunction might be the key clinical features that determine ACLF prognosis.

To identify ACLF prognostic factors, we validated available liver prognostic scoring models, which were developed from various populations, and evaluated independent clinical parameters based on our population. Univariate analysis showed that creatinine, INR, bilirubin, lactate, bicarbonate, albumin, ACLF grading and prognostic models were parameters associated with short-term mortality rates. From these independent parameters, particular prognostic models, including CTP, ALBI, MELD and MELD-Na scores, might have multicollinearity with some clinical parameters, such as bilirubin, creatinine, INR, and albumin. Therefore, we did not include these models in the multivariate regression analysis for two reasons: 1) these scores had a milder degree of association than individual clinical parameters; and 2) their predictive accuracies were poor (AUROC below 0.6), so we chose only CLIF-C OF and CLIF-C ACLF scores as representative prognostic scores in multivariate regression analysis. It should be noted that although the ALBI score was found to be associated with 30-day mortality with a relatively high degree of association, a separate multivariate analysis (S2 Table) among prognostic scores showed non-significantly associated with mortality. In addition, the ALBI score was initially designed to predict mortality in liver cancer. There has been few evidence to support the use of ALBI score in ACLF [23–25], we therefore decided not to include the ALBI score in the analysis with other clinical parameters. Multivariate analysis showed that creatinine and the CLIF-C OF score remained statistically significant for 30-day mortality prediction and the CLIF-C OF score remained the only predictor for 90-day mortality.

A previous study found that the CLIF-SOFA score is valid [7]; however, modified scores such as the CLIF-C OF score and CLIF-C ACLF score had not been studied. The current study found that ACLF-specific scores, including the CLIF-C ACLF and CLIF-OF scores, were superior to other general prognostic models, such as the CTP score, MELD score and MELD-Na score, in predicting short-term mortality. These outperformance trends of ACLF-specific scores were similar to those in a previous study from Thailand [3], as well as other studies from Asia [26–29] and multicenter studies from the EASL working group [30]. In addition to

prognostic scores, some parameters indicating the degree of organ dysfunction and ACLF severity were identified as independent prognostic predictive parameters. Multivariate regression analysis suggested that the CLIF-OF score was the best prognostic predictor that could predict both 30-day and 90-day mortality, followed by creatinine, which predicted only 30-day mortality well. Because the CLIF-OF score is composed of both clinical parameters and laboratory profiles, this might explain the better performance in predicting prognosis over using a single laboratory profile alone.

The mortality rate in AD patients was higher than that in other studies (25.5% vs. 3.39%-10% and 32.78% vs. 5.33%-24% at 30 days and 90 days, respectively) [3, 28, 31–34]. It should be noted that the 30-day mortality rate in AD patients from our study was greater than 15%, which was a threshold definition of high short-term mortality for ACLF [2]. Recently, the PREDICT study [35], which was led by researchers from the EASL, has identified three distinct clinical courses in AD patients: 1) pre-ACLF, 2) unstable decompensated cirrhosis (UDC), and 3) stable decompensated cirrhosis (SDC). It is possible that the majority of AD patients in our study had pre-ACLF or UDC rather than SDC, which might be supported by findings that the levels of white blood cells, bilirubin, INR, albumin, creatinine, CTP score and MELD score in our study were close to the average level in the pre-ACLF and UDC groups from the PRE-DICT study [35]. However, the exact cause of the extremely high mortality rates in our population remains unclear.

Regarding prognostic predictors for AD patients, none of the biochemical profiles, prognostic scores, or acute liver insult types were significantly associated with 28-day mortality. The only potential 28-day prognostic predictor was HE. Although its p value was not small enough to allow us to claim statistical significance (p value = 0.055), the degree of association with an OR of 2.24 made it worth commenting on. In addition, its clinical importance was supported by findings from various studies [36–38]. Further study with a larger sample size may be needed. For 90-day prognosis, bacterial infection was an independent factor associated with mortality. Previously, the PREDICT study hypothesized that systemic inflammation and severe portal hypertension played major roles in determining the clinical course of AD, especially in the pre-ACLF and UDC groups [35, 39]. Our study found that bacterial infection, a potent trigger of the systemic inflammatory response, was a poor 90-day prognostic predictor; however, none of the other biochemical inflammatory parameters could predict mortality. Therefore, although systemic inflammation is a dominant determining factor for AD prognosis, there might be other potential mechanisms contributing to AD death, such as impaired neurohormonal response to liver injury, impairment of control in molecular biology response after tissue injury, for instance, loss of control of cellular metabolism (glucose, lipids and proteins), mitochondrial dysfunction, and increased reactive oxygen species [40]. These alterations in cellular metabolism might explain the variation in the clinical course of AD patients who had a similar phenotype of baseline inflammation but had different prognoses.

In addition to bacterial infection, the CLIF-AD score and CTP score were found to be potential parameters that correlated with 90-day mortality. However, their predictive accuracies were unsatisfactory, and even the CLIF-C AD score, which was excellent according to the EASL working group [33] and has been validated by several studies [31, 32, 41, 42], became statistically non-significant after multivariate analysis. As there are no current initial parameters that could adequately predict prognosis in severe AD patients, novel biomarkers to detect impairment of cellular metabolism early are needed. In the real-world setting, another practical strategy is to perform serial prognostic assessments within a couple of days after admission, which might predict clinical outcome more accurately than using only parameters on the day of admission.

This study has some limitations. First, all study centers were university hospitals; thus, our study was subjected to referral bias and had a higher proportion of severe cases. These factors

possibly contributed to a worse prognosis than that of the general population. In addition, a large proportion of participants in our study were referred from primary or secondary hospitals, and some participants received partial treatment before referral. It is uncertain whether the delay prognostic score calculation or receiving partially treatment might had some influence to accuracies of prognostic scores. Second, because we enrolled patients from 2015 until 2020, a long duration of study enrollment might result in heterogeneity of medical practices that could change following the rapid growing knowledge in recent years. Third, as the concept of different distinct clinical courses of AD has been proposed after we had completed study enrollment, we could not classify subtypes of patients, and prognostic assessment in each subtype could not be performed. For future research, we plan to include more participants from various levels of hospital capacity and develop prognostic scores in each subgroup of AD patients.

## Conclusions

Cirrhosis patients who were hospitalized due to acute liver decompensation with or without ACLF had a high short-term mortality rate. The CLIF-C OF score demonstrated the best accuracy in predicting the prognosis of ACLF patients. The prognosis of AD patients was poorer than that of other populations, and appropriate parameters that could accurately predict mortality were not found. Further research is needed to solve these challenges.

## Supporting information

**S1 Table. Diagnostic performance of each prognostic score in predicting short-term mortality.**
(DOCX)

**S2 Table. Predictive performance of prognostic scores for mortality among patients with ACLF.**
(DOCX)

**S3 Table. Predictive factors for mortality among patients with ACLF.**
(DOCX)

**S4 Table. Predictive factors for mortality among patients with AD without ACLF.**
(DOCX)

**S1 File. Data sharing document.**
(XLSX)

## Acknowledgments

This manuscript was supported for English editing by the research team of the Department of Medicine, Faculty of Medicine, Chulalongkorn University.

## Author Contributions

**Conceptualization:** Tongluk Teerasarntipan, Kessarin Thanapirom, Sombat Treeprasertsuk.

**Data curation:** Sakkarin Chirapongsathorn, Tanita Suttichaimongkol, Naichaya Chamroonkul, Chalermrat Bunchorntavakul, Sith Siramolpiwat, Siwaporn Chainuvati, Abhasnee Sobhonslidsuk, Apinya Leerapun, Teerha Piratvisuth, Wattana Sukeepaisarnjaroen, Tawesak Tanwandee, Sombat Treeprasertsuk.

**Formal analysis:** Tongluk Teerasarntipan, Kessarin Thanapirom.

**Funding acquisition:** Sombat Treeprasertsuk.

**Investigation:** Tongluk Teerasarntipan, Sakkarin Chirapongsathorn, Tanita Suttichaimong-kol, Naichaya Chamroonkul, Chalermrat Bunchorntavakul, Sith Siramolpiwat, Siwaporn Chainuvati, Abhasnee Sobhonslidsuk, Apinya Leerapun, Teerha Piratvisuth, Wattana Sukeepaisarnjaroen, Tawesak Tanwandee.

**Methodology:** Tongluk Teerasarntipan.

**Project administration:** Sombat Treeprasertsuk.

**Supervision:** Sombat Treeprasertsuk.

**Validation:** Sakkarin Chirapongsathorn, Sombat Treeprasertsuk.

**Writing – original draft:** Tongluk Teerasarntipan, Kessarin Thanapirom.

**Writing – review & editing:** Sakkarin Chirapongsathorn, Tanita Suttichaimongkol, Naichaya Chamroonkul, Chalermrat Bunchorntavakul, Sith Siramolpiwat, Siwaporn Chainuvati, Abhasnee Sobhonslidsuk, Apinya Leerapun, Teerha Piratvisuth, Wattana Sukeepaisarnjaroen, Tawesak Tanwandee, Sombat Treeprasertsuk.

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
