## [Decision Letter · Decision Letter 0]

6 Sep 2022

PONE-D-22-22364Validation of prognostic scores predicting mortality in acute liver decompensation or acute-on-chronic liver failure: a Thailand multicenter studyPLOS ONE

Dear Dr. Treeprasertsuk,

Thank you for submitting your manuscript to PLOS ONE. After careful consideration, we feel that it has merit but does not fully meet PLOS ONE’s publication criteria as it currently stands. Therefore, we invite you to submit a revised version of the manuscript that addresses the points raised during the review process.

As you can see, both reviewers appreciated that this is a large, relatively well study, but also requested important changes that need to be done before the manuscript can be considered further.

We look forward to receiving your revised manuscript.

Kind regards,

Pavel Strnad

Academic Editor

PLOS ONE

Reviewers' comments:

Reviewer's Responses to Questions

**Comments to the Author**

1. Is the manuscript technically sound, and do the data support the conclusions?

Reviewer #1: Yes

Reviewer #2: Yes

2. Has the statistical analysis been performed appropriately and rigorously? 

Reviewer #1: Yes

Reviewer #2: I Don't Know

3. Have the authors made all data underlying the findings in their manuscript fully available?

Reviewer #1: Yes

Reviewer #2: Yes

4. Is the manuscript presented in an intelligible fashion and written in standard English?

Reviewer #1: Yes

Reviewer #2: Yes

5. Review Comments to the Author

Reviewer #1: The authors of a manuscript “PONE-D-22-22364: Validation of prognostic scores predicting mortality in acute liver decompensation or acute-on-chronic liver failure: a Thailand multicenter study” present a retrospective observational study on cirrhotic patients with acute decompensation of cirrhosis and acute-on-chronic liver failure (ACLF) in Thailand. The authors aimed to describe prognostic factors for mortality in Asian population. The authors conclude that for ACLF patients the best predictor of 30-day and 90-day mortality is CLIF-C OF score, while for acute decompensation of cirrhosis the best predictor for 90-day mortality are bacterial infection, Child-Pugh score and CLIF-AD score (but without the sufficient significance).

Comments:

1. Though the paper is written in good English, some sentences are not clear (for example the first sentence in the Background in Abstract).

2. In the Abstract the authors write that the study was retrospective. In “Study design” the description of prospective data collection is mentioned. How many patients were evaluated prospectively?

3. In Statistical analysis the authors write that total of 374 participants were required. For what correlation or analysis was that number of patients required??

4. Six patients in ACLF group were transplanted. Were also patients with decompensation of cirrhosis indicated for liver transplantation? Was the artificial liver support available for ACLF patients?

5. MELD score was not related to the mortality in patients with cirrhosis decompensation. Do the authors have an explanation?

6. Do the authors have an information about the cause of death in patients in the group with liver decompensation? Could some patients develop ACLD during the period of observation?

This is an interesting study describing the usefulness of CLIF prognostic scores both in patients with ACLF and in patients with acute decompensation of cirrhosis without organ failure in Asian population.

Reviewer #2: I have read the article of Teerasarntipan et al “Validation of prognostic scores predicting mortality in acute liver decompensation or acute-on-chronic liver failure: a Thailand multicenter study.

It is a validation study of the CLIF-AD and CLIF-C OF score in a large Thai population of ACLF patients, and also identifies independent predictors of short-term mortality. It is a large study with adequate rationale and conclusions, but it is not the first validation study of these prognostic scores in an Asian population as the authors also mention.

I have some questions regarding the design of the study:

The authors have 2 separate research questions in this manuscript: 1. The validation of the CLIF-C OF/ClLIF-C ACLF and CLIF-AD score in a Thai AD/ACLF population and 2. The identification of prognostic factors for 30-day and 90-day mortality in AD +/- ACLF patients. They combined these research questions by performing multivariate analyses including separate variables and (some) of the scores. However, the CLIF-C OF score and CLIF-C ACLF score highly overlap and also include variables that are also separately included (age, WBC, bilirubin, INR). It is better to exclude the prognostic scores from the multivariate analyses and just evaluate their performance in the current population by AUROC. The identification of prognostic factors by multivariate analysis should be a separate analysis and could give some insight why standard prognostic scores underperform in this Thai population. The authors should also elaborate why the standard prognostic scores underperform in their study, and whether there is a need for other prognostic scores in Thai populations.

The authors mention the high 30- and 90-day mortality rate observed in AD patients. They mention in the methods that all admitted AD patient were scores as ACLF patients in case they developed organ failure and classified for ACLF during the admission. Did these patients die without developing ACLF, or after the admission? How were patient classified when they were readmitted due to ACLF after an admission due to AD?

This is one of the many validation studies of the CLIF-C ACLF and CLIF-AD score. What is th

It is not clear to me why there were exactly 301 patients in each group (AD and ACLF). This suggest that sort of matching was performed, but this is not clear from the statistics section.

How were bacterial infections defined?

6. PLOS authors have the option to publish the peer review history of their article (what does this mean?). If published, this will include your full peer review and any attached files.

Reviewer #1: No

Reviewer #2: No

---

## [Author Response · Author response to Decision Letter 0]

26 Oct 2022

Reviewers’ comments:

Reviewer #1

Comment 1. Though the paper is written in good English, some sentences are not clear (for example the first sentence in the Background in Abstract).

Response: We thank the reviewer for the comments. We revised some words in the Background of abstract for the better understanding as follows: 

“Cirrhosis patients with worsening of the liver function are defined as acute decompensation (AD) and those who develop extrahepatic organ failure are defined as acute-on-chronic liver failure (ACLF). Both AD and ACLF have an extremely poor prognosis.” (Page 3, line 48-50)

Comment 2. In the Abstract the authors write that the study was retrospective. In “Study design” the description of prospective data collection is mentioned. How many patients were evaluated prospectively?

Response: From 602 participants, 48 patients (7.9%) were retrospectively recruited after AD or ACLF occurred but the 30-day and 90-day evaluation, we prospectively collected their data at 30 days and 90 days after the onset of liver decompensation events. We have added the following sentence in the Results section:

“There were 48 participants (7.9%) retrospectively recruited after AD or ACLF occurred but the 30-day and 90-day evaluation, we prospectively collected their data after the onset of liver decompensation events.” (Page 8, Line 194-196)

Comment 3. In Statistical analysis the authors write that total of 374 participants were required. For what correlation or analysis was that number of patients required??

Response: To estimate an infinite population proportion, we considered a mortality of 22.1% based on the ACLF grade 1 mortality rate from the CANONIC study with a power of 80% and 0.05 type 1 error, including 10% lost to follow-up rate, a total 374 participants were required. We have revised these phrases in the Statistical Analysis section as follow: 

“The sample size of calculated to estimate an infinite population proportion considering a mortality of 22.1% based on the ACLF grade 1 mortality rate from the CANONIC study[2] and it was found that a minimum of 340 patients were required to achieve this observational result with a power of 80 percent and 0.05 type 1 error. With additional 10% lost to follow-up rate, a total of 374 participants were required in this study.” (Page 7, Line 165-169)

Comment 4. Six patients in ACLF group were transplanted. Were also patients with decompensation of cirrhosis indicated for liver transplantation? Was the artificial liver support available for ACLF patients?

Response: All participants who underwent liver transplanted were already on waiting list with an indication of chronic liver decompensation before the onset of ACLF in this study. ACLF may resulted in the higher priority of receiving liver transplantation in few cases. Artificial liver support was not available for ACLF patients in our study. We have added this information in the Result section as follow: 

“Six patients (1.99%), who were currently on the waiting list before the onset of ACLF, underwent liver transplantation within 90 days after ACLF diagnosis.” (Page 11, Line 221)

Comment 5. MELD score was not related to the mortality in patients with cirrhosis decompensation. Do the authors have an explanation?

Response: It should be noted that its statistically significant at 90-day period reached borderline, we might infer that MELD score have potential to be useful to predict mortality at the longer period after AD events, such as 6 months or 12 months. However, it is unclear why MELD score had unsatisfying prediction value in short-term mortality. 

Comment 6. Do the authors have an information about the cause of death in patients in the group with liver decompensation? Could some patients develop ACLD during the period of observation?

Response: We thank the reviewer for the comment. Regarding the development of ACLF during the period of observation, patients who presented with AD features and developed ACLF during admission were categorized into the ACLF group. (We have mentioned this in the Study design section, page, line). Regarding the cause of death and post-admission ACLF, the mortality rate in AD patients was much higher than we expected, so we did not collect information about the cause of death in AD patients and the occurrence of ACLF after discharge in the research protocol and ethics proposal. Unfortunately, with some ethical limitation, we were not allowed to retrospectively search these issues in the electronic medical records. 

Reviewer #2: 

Comments 1: The authors have 2 separate research questions in this manuscript: 1. The validation of the CLIF-C OF/ClLIF-C ACLF and CLIF-AD score in a Thai AD/ACLF population and 2. The identification of prognostic factors for 30-day and 90-day mortality in AD +/- ACLF patients. They combined these research questions by performing multivariate analyses including separate variables and (some) of the scores. However, the CLIF-C OF score and CLIF-C ACLF score highly overlap and also include variables that are also separately included (age, WBC, bilirubin, INR). It is better to exclude the prognostic scores from the multivariate analyses and just evaluate their performance in the current population by AUROC. 

Response: We thank the reviewer very much for the comments. We concerned of the overlapping variables in the CLIF-C OF score and CLIF-C ACLF score. We evaluated their performance separately by AUROC and described them in the Results section of the main text, Page 11, Line 241 to Page 12, Line 246. We also demonstrated this information in the submitted Figure 3 and S1 Table.

Comment 2: The identification of prognostic factors by multivariate analysis should be a separate analysis and could give some insight why standard prognostic scores underperform in this Thai population.

Response: We thank the reviewer for valuable suggestion. Indeed, we have already performed a multivariate analysis using separate prognostic scores. 

We found that the results were relatively similar to results from the multivariate analysis that included other parameters as we shown in the main text. By separate analysis, CLIF-C OF score was the best predictor for 30-day and 90-day mortality among ACLF patients with adjusted odd ratio of 1.30 and 1.23, respectively, in comparison with adjusted odd ratio of 1.44 and 1.48, respectively, that were found by analyses with other variables. We think that some other parameters that were not included in the prognostic scores might have influence to mortality, such as ACLF grading or serum lactate level, therefore a combination of prognostic scores and other clinical parameters would be the most comprehensive evaluation for mortality prediction. 

Comment 3: Whether there is a need for other prognostic scores in Thai populations?

Response: The Albumin-Bilirubin (ALBI) score is currently shown the evidence of good index for liver function evaluation and prognosis evaluation in patients with hepatocellular carcinoma patients receiving TACE. Recently, there are some studies applied the Albumin-Bilirubin Score in patients with ACLF. 

In this revision, we calculated prognostic performance of the ALBI score, combined with other standard prognostic scores, to predict mortality in our patients. The revised separate analysis of prognostic scores were shown in the S2 Table. 

By univariate analysis, ALBI score was found to be associated with only 30-day mortality (p<0.001) with fair accuracy (AUROC of 0.605, p=0.002). While it was not associated with 90-day mortality (p=0.08) with an unsatisfactory accuracy (AUROC of 0.568, p=0.078). After adjusted by multivariate analysis, its predictive performance become non-significant, while predictive performance of other standard prognostic scores was remained in the same trends with previous analysis. We think that there has been insufficient evident to support the use of ALBI score to predict mortality in ACLF patients, we therefore decided not to include the ALBI score in the overall analysis. 

We added this revised table in the supplemental documents as S2 Table (Page 12, Line 261) and updated the sequence number of each supplemental table (S3-S4 Tables). 

We added information regarding the prognostic value of the ALBI scores in the Table 3. (Page 14, Line 269) 

We added results of the separate analysis of prognostic scores in the Results section as follow:

“Multivariate analysis of standard prognostic scores found that CLIF-C OF scores were associated with 30-day and 90-day mortality, while CLIF-C ACLF scores was associated with 30-day mortality. (S2 table) Subsequently, we performed multivariate analysis combining clinical parameters with selected prognostic scores.” (Page 12, Line 259-262)

We added discussion regarding the use of ALBI score and its limitation in the Discussion section as follow:

“It should be noted that although the ALBI score was found to be associated with 30-day mortality with a relatively high degree of association, a separate multivariate analysis (S2 table) among prognostic scores showed non-significantly associated with mortality. In addition, the ALBI score was initially designed to predict mortality in liver cancer. There has been few evidence to support the use of ALBI score in ACLF[23-25], we therefore decided not to include the ALBI score in the analysis with other clinical parameters.” (Page 16, Line 329 to Page 17, Line 334)

Comment 4: The authors should also elaborate why the standard prognostic scores underperform in their study.

Response: It is unclear that why the standard prognostic scores in our study were slightly underperform in comparison to other studies. It might be observed that all study centers were university hospitals. A large proportion of participants were referred from primary or secondary hospitals. It is uncertain whether the delay prognostic score calculation, such as at the first day of liver injury versus the later day, contributed to the underperformance of prognostic scores. In addition, some patients had been partially treated before referral to research centers, which might be a confounder for diagnostic accuracies in the performance of prognostic scores. 

We added this discussion in the study limitation part of the Discussion section as follow:

“In addition, a large proportion of participants in our study were referred from primary or secondary hospitals, and some participants received partial treatment before referral. It is uncertain whether the delay prognostic score calculation or receiving partially treatment might had some influence to accuracies of prognostic scores.” (Page 19, Line 390-393)

Comment 5: The authors mention the high 30- and 90-day mortality rate observed in AD patients. They mention in the methods that all admitted AD patient were scores as ACLF patients in case they developed organ failure and classified for ACLF during the admission. Did these patients die without developing ACLF, or after the admission? How were patient classified when they were readmitted due to ACLF after an admission due to AD?

Response: In this study, a number of AD patients died without developing ACLF, which occurred either during admission or after admission. For AD patients who are readmitted and develop ACLF, we reclassified them into ACLF group and reevaluate their prognosis at their last admission. 

Comment 6: It is not clear to me why there were exactly 301 patients in each group (AD and ACLF). This suggest that sort of matching was performed, but this is not clear from the statistics section.

Response: The number of participants in each group were exactly 301 patients by chance. We did not perform matching or using any methodological adjustment. 

Comment 7: How were bacterial infections defined?

Response: Bacterial infection was defined by one of the following conditions: 1) detection of positive bacterial culture in blood or any clinically suspicious organ 2) clinically and laboratory compatible with sepsis or spontaneous bacterial peritonitis with-or without of microbiologically documented infection. 

We added these phrases in the Methods section in the main text as follow:

“We defined bacterial infection by one of the following conditions: 1) detection of positive bacterial culture in blood or any clinically suspicious organ 2) clinically and laboratory compatible with sepsis or spontaneous bacterial peritonitis with-or without of microbiologically documented infection.” (Page 6, Line 135-138)

We believe that the above-mentioned efforts have significantly improved this revised manuscript and hope that it is acceptable for publication in PLOS ONE. Thank you in advance for your efforts on this revised manuscript

---

## [Decision Letter · Decision Letter 1]

8 Nov 2022

Validation of prognostic scores predicting mortality in acute liver decompensation or acute-on-chronic liver failure: a Thailand multicenter study

PONE-D-22-22364R1

Dear Dr. Treeprasertsuk,

We’re pleased to inform you that your manuscript has been judged scientifically suitable for publication and will be formally accepted for publication once it meets all outstanding technical requirements.

Kind regards,

Pavel Strnad

Academic Editor

PLOS ONE

Additional Editor Comments (optional): Congratulations to the nice work!

Reviewers' comments:

Reviewer's Responses to Questions

**Comments to the Author**

1. If the authors have adequately addressed your comments raised in a previous round of review and you feel that this manuscript is now acceptable for publication, you may indicate that here to bypass the “Comments to the Author” section, enter your conflict of interest statement in the “Confidential to Editor” section, and submit your "Accept" recommendation.

Reviewer #1: All comments have been addressed

Reviewer #2: All comments have been addressed

2. Is the manuscript technically sound, and do the data support the conclusions?

Reviewer #1: (No Response)

Reviewer #2: Yes

3. Has the statistical analysis been performed appropriately and rigorously? 

Reviewer #1: (No Response)

Reviewer #2: Yes

4. Have the authors made all data underlying the findings in their manuscript fully available?

Reviewer #1: (No Response)

Reviewer #2: Yes

5. Is the manuscript presented in an intelligible fashion and written in standard English?

Reviewer #1: (No Response)

Reviewer #2: Yes

6. Review Comments to the Author

Reviewer #1: (No Response)

Reviewer #2: (No Response)

7. PLOS authors have the option to publish the peer review history of their article (what does this mean?). If published, this will include your full peer review and any attached files.

Reviewer #1: No

Reviewer #2: No

---

## [Editor Report · Acceptance letter]

14 Nov 2022

PONE-D-22-22364R1 

Validation of prognostic scores predicting mortality in acute liver decompensation or acute-on-chronic liver failure: a Thailand multicenter study 

Dear Dr. Treeprasertsuk:

I'm pleased to inform you that your manuscript has been deemed suitable for publication in PLOS ONE. Congratulations! Your manuscript is now with our production department. 

Kind regards, 

on behalf of

Dr. Pavel Strnad 

Academic Editor

PLOS ONE